# Beyond wind-induced upwelling: diverse drivers of future productivity in eastern boundary upwelling systems

Erica Cioffi<sup>1</sup>, Laurent Bopp<sup>1</sup>, and Lester Kwiatkowski<sup>2</sup>

<sup>1</sup>Laboratoire de Météorologie Dynamique/Institut Pierre-Simon Laplace, Ecole Normale Supérieure/Université Paris Sciences et Lettres, Département de Géosciences/CNRS/Ecole Polytechnique/Sorbonne Université, Paris, 75005, France <sup>2</sup>Laboratoire d'océanographie et du climat: expérimentations et approches numériques/Institut Pierre-Simon Laplace, Sorbonne Université/CNRS/Institut de recherche pour le développement/Muséum national d'Histoire naturelle, Paris, 75005, France

**Correspondence:** Erica Cioffi (erica.cioffi@lmd.ipsl.fr)

Abstract. Eastern Boundary Upwelling Systems (EBUS) contribute disproportionately to global marine productivity and fisheries, yet their response to climate change remains poorly understood. Given the essential ecosystem services they support, improving projections of future EBUS dynamics is critical. Here we analyze projections of Net Primary Production (NPP) and its driving mechanisms using Earth System Models (ESMs) from the Coupled Model Intercomparison Project Phase 6 (CMIP6). Across the four major EBUS, twenty-first century NPP projections exhibit larger model uncertainty than scenario uncertainty, with limited confidence in the direction of future trends under different scenarios. This uncertainty partially results from compensating positive and negative NPP anomalies within individual systems, with consistent multi-model responses only emerging at subsystem scales. Although, consistent with most past studies, changes in upwelling-favorable winds are an important driver of the EBUS NPP response to climate change, they cannot fully explain projected responses. In the equatorward sectors of the Canary and Benguela systems, as well as in the historically most productive area of the California system (regions encapsulating 25 % of total EBUS area) a weakening of alongshore wind stress reduces upwelling intensity, nutrient supply to the euphotic zone and consequently NPP. However, in the remaining 75 % of EBUS extent, additional mechanisms are required to explain projected changes. These include upwelling anomalies induced by geostrophic transport and windstress curl, enhanced stratification, and changes in subsurface nutrient reservoirs, highlighting the complex and locally-specific response of EBUS productivity to climate change.

#### 1 Introduction

The major Eastern Boundary Upwelling Systems (EBUS), including the California (CalCS), Canary (CanCS), Humboldt (HumCS), and Benguela (BenCS) Current Systems, are among the most biologically productive regions of the global ocean. Although they cover only  $\sim 1\%$  of the ocean surface, their contribution to global marine productivity is disproportionately large ( $\sim 7\%$ ) (Pauly and Christensen, 1995), Fig. 1. The phytoplankton in these regions play a key role in the carbon cycle by fixing carbon dioxide through photosynthesis (Stukel et al., 2023) and acting as the base of the marine food chain, sustaining rich biodiversity of fish, seabirds, and marine mammals (Chavez and Messié, 2009). EBUS also support large marine resources,

25

45

accounting for up to 20 % of the global fish catch (Pauly and Christensen, 1995) and providing economic and social services to about 80 million people (García-Reyes et al., 2015).

EBUS primary production is largely driven by the high availability of macro- and micronutrients in the euphotic zone. These nutrients are supplied through intense coastal upwelling to surface layers, where they become accessible to phytoplankton. This process is primarily associated with offshore Ekman transport, induced by the alongshore equatorward winds that result from the atmospheric pressure gradient between high-pressure systems over subtropical ocean basins and low-pressure systems over adjacent land masses. In addition to this dominant mechanism, other physical processes also influence the supply of nutrients, such as cross-shore geostrophic transport near the coast (Oerder et al., 2015; Jacox et al., 2018) or Ekman pumping driven by wind-stress curl farther offshore (Bravo et al., 2016; Jacox et al., 2014).

Historically, perturbations to wind-induced upwelling have been thought to be the dominant drivers of the EBUS phytoplankton NPP response to climate change (Bakun, 1990; Sydeman et al., 2014; Bakun et al., 2015). Yet, projections of NPP remain uncertain, due to the interplay of multiple influencing factors, which can either reinforce or offset each other over varying time scales (Bograd et al., 2023; García-Reyes et al., 2015).

Most past studies have focused on how future variations in alongshore wind-induced upwelling will affect NPP (Bakun, 1990; Rykaczewski and Checkley Jr, 2008; Wang et al., 2015). Bakun's early theory (Bakun, 1990) suggested that global warming would enhance the ocean-land temperature and pressure gradient, strengthening coastal winds, intensifying upwelling, and increasing NPP. More recent research (Rykaczewski et al., 2015; Bograd et al., 2023) has extended this hypothesis, identifying a latitude-dependent NPP response within EBUS due to the potential future displacement of major high-pressure systems toward higher latitudes. This shift, associated with the expansion of the Hadley cell (Lu et al., 2007; Grise and Davis, 2020) and the increase in the Southern Annular Mode (Gillett and Fyfe, 2013), could influence summertime alongshore winds, which are projected to intensify in the poleward regions of the EBUS and to weaken in the equatorward regions, potentially affecting the intensity and location of upwellings.

Such changes to large-scale atmospheric pressure systems may also affect the wind-stress curl in the EBUS, influencing the characteristics of induced upwelling (Pickett and Paduan, 2003; Rykaczewski and Checkley Jr, 2008). Beyond atmospheric drivers, future upwelling has also been shown to be modulated by shifts in geostrophic transport, associated with variations in sea surface height linked to global warming, which may offset the role of Ekman transport anomalies in some EBUS regions (Ding et al., 2021; Jing et al., 2023). In addition to changes to the physical drivers of upwelling at depth, nutrient supply to the surface is further influenced by increased local stratification, which can limit transport efficiency, both by weakening mixing across the pycnocline and by causing a shallower source of upwelled waters (Roemmich and McGowan, 1995; Jacox and Edwards, 2011; Sousa et al., 2020).

Alongside projected changes in vertical velocity, shifts in the timing and duration of the upwelling season have been identified as potential contributors to total upwelling intensity, with future coastal upwelling trends showing heterogeneity throughout the year (Vázquez et al., 2023; Sousa et al., 2017). Earlier onset and prolonged duration of upwelling seasons are anticipated in the EBUS, attributed to variations in Ekman and geostrophic transport in the Pacific and Atlantic EBUS, respectively (Du et al., 2024). However, such changes have been found to have limited impact on total upwelling intensity compared to changes

in vertical velocity.

Variations in the physical drivers of circulation and mixing, together with changes in the biogeochemical properties of source waters, modulate the nutrient supply to EBUS. Shifts in vertical nutrient transport have been shown to correlate well with changes in NPP (Du et al., 2024), with alterations in water column nitrate concentrations identified as critical in shaping future primary production (Jacox et al., 2024; Messié et al., 2009; Bristow et al., 2017; Jacox et al., 2016). Nitrate enrichment of upwelled waters, driven by changes in local processes (such as increased remineralization (Dussin et al., 2019; Rykaczewski and Dunne, 2010)) or basin-scale circulation (e.g. decreased ventilation at high-latitudes) has been shown to enhance productivity, compensating for changes in upwelling intensity and stratification (Rykaczewski and Dunne, 2010; Pozo Buil et al., 2021; Howard et al., 2020).

At present, the limited duration of observational records and associated measurement uncertainty hinder a robust assessment of climate impacts on EBUS NPP (Henson et al., 2010). Some studies have reported evidence of productivity trends over recent decades, with increases in certain regions and declines elsewhere (Kahru et al., 2009; Arístegui et al., 2009; Lamont et al., 2019; Weidberg et al., 2020). However, these findings often differ depending on the data source, and consistent long-term trends have not been established. For example, there is only medium evidence and medium agreement that primary production in the Canary Current has decreased over the past 2 decades (Arístegui et al., 2009; Demarcq, 2009). To address the limitations of observational records and understand potential changes of primary production in these systems, climate model simulations are increasingly employed.

Here we reevaluate the standard perspective that perturbations to upwelling favorable winds determine the EBUS NPP response to climate change. Using ESM simulations conducted within CMIP6 (O'Neill et al., 2016; Eyring et al., 2016) we assess the projected twenty-first century evolution of NPP across the four major EBUS under different Shared Socioeconomic Pathways (SSPs) (Riahi et al., 2017; Meinshausen et al., 2020). Specifically, we evaluate the extent to which projected changes in upwelling favorable winds are consistent with changes in vertical seawater velocities, upper ocean nutrient concentrations and NPP. The multi-model robustness of projections is evaluated and where NPP anomalies cannot be mechanistically explained by perturbations to upwelling-favorable winds, alternative drivers are explored, including changes to subsurface nitrate reservoirs and increasing ocean stratification.

#### 2 Materials and methods

A multi-model ensemble approach was used to investigate the mechanisms driving changes in EBUS NPP under climate change. CMIP6 ESMs were analyzed, with the baseline period of historical simulations (1985–2014) compared with twenty-first century SSP projections (2015–2100).

## 2.1 Models and data

Model outputs were obtained from the CMIP6 archive hosted by the Earth System Grid Federation (ESGF). The selected ESMs simulate coupled physical (atmosphere, land, ocean, and sea ice) and biogeochemical processes (Table 1), allowing for

a consistent evaluation of NPP changes and their potential drivers. The database underwent homogenization using distance-weighted average remapping: the resulting regridded files share a common horizontal resolution of 1 degree (360 × 180), while the vertical grid for each model retains its native discretization, which ranges from 40 to 75 levels. For each model, a single ensemble member was considered for each experiment (typically r1i1p1f1, sometimes r1i1p1f2 or r1i2p1f1), hence the potential influence of internal variability on projected changes was not considered.

The evaluated scenarios were SSP1-2.6, SSP2-4.5, SSP3-7.0, and SSP5-8.5, with the latter specifically used to analyze the mechanisms driving climate-related NPP impacts, as the magnitude of projected changes is larger than in lower-emissions scenarios. While 18 CMIP6 models were used to assess changes in NPP, a subset of 13 models was selected for the full mechanistic analysis. This subset was based on the availability of complete datasets for all relevant variables over both historical and projection periods, ensuring internal consistency while maintaining ensemble diversity.

**Table 1:** The CMIP6 Earth System Models used, their atmospheric, ocean, and marine biogeochemistry components with the relative resolutions  $^{a}$ . The last column lists the experiments assessed for NPP projections (red circle for SSP5-8.5, yellow for SSP3-7.0, blue for SSP2-4.5, green for SSP1-2.6). For SSP5-8.5, it is indicated whether a model is part of the ensemble subset used for full mechanistic analysis ( $\checkmark$ ) or not (X).

| Atmosphere                            | Ocean Biogeochemistry                                                                                                                                                                            |                                                                                                                                                                                                                                                                                                                                                                                                                                                                                                                                                                                                                                                                                                                                                                                                        | Availability                                                                                                                                                                                                                                                                                                                                                                                                             |
|---------------------------------------|--------------------------------------------------------------------------------------------------------------------------------------------------------------------------------------------------|--------------------------------------------------------------------------------------------------------------------------------------------------------------------------------------------------------------------------------------------------------------------------------------------------------------------------------------------------------------------------------------------------------------------------------------------------------------------------------------------------------------------------------------------------------------------------------------------------------------------------------------------------------------------------------------------------------------------------------------------------------------------------------------------------------|--------------------------------------------------------------------------------------------------------------------------------------------------------------------------------------------------------------------------------------------------------------------------------------------------------------------------------------------------------------------------------------------------------------------------|
| UM7.3 Approx. GA1                     | MOM5 WOMBAT                                                                                                                                                                                      |                                                                                                                                                                                                                                                                                                                                                                                                                                                                                                                                                                                                                                                                                                                                                                                                        | • • • •                                                                                                                                                                                                                                                                                                                                                                                                                  |
| $1.8758^{\circ} \times 1.258^{\circ}$ | ~1° × 1° b                                                                                                                                                                                       |                                                                                                                                                                                                                                                                                                                                                                                                                                                                                                                                                                                                                                                                                                                                                                                                        | $\checkmark$                                                                                                                                                                                                                                                                                                                                                                                                             |
| CanAM5                                | NEMOv3.4.1                                                                                                                                                                                       | CMOC                                                                                                                                                                                                                                                                                                                                                                                                                                                                                                                                                                                                                                                                                                                                                                                                   | • • • •                                                                                                                                                                                                                                                                                                                                                                                                                  |
| T63 ( $\sim$ 2.8° $\times$ 2.8°)      | ~1° × 1° °                                                                                                                                                                                       |                                                                                                                                                                                                                                                                                                                                                                                                                                                                                                                                                                                                                                                                                                                                                                                                        | $\checkmark$                                                                                                                                                                                                                                                                                                                                                                                                             |
| CanAM5                                | NEMOv3.4.1                                                                                                                                                                                       | CanOE                                                                                                                                                                                                                                                                                                                                                                                                                                                                                                                                                                                                                                                                                                                                                                                                  | • • • •                                                                                                                                                                                                                                                                                                                                                                                                                  |
| T63 ( $\sim$ 2.8° $\times$ 2.8°)      | ~1° × 1° °                                                                                                                                                                                       |                                                                                                                                                                                                                                                                                                                                                                                                                                                                                                                                                                                                                                                                                                                                                                                                        | $\checkmark$                                                                                                                                                                                                                                                                                                                                                                                                             |
| CAM6                                  | POP2                                                                                                                                                                                             | MARBL-BEC                                                                                                                                                                                                                                                                                                                                                                                                                                                                                                                                                                                                                                                                                                                                                                                              | • • • •                                                                                                                                                                                                                                                                                                                                                                                                                  |
| $1.25^{\circ} \times 0.9^{\circ}$     | $\sim$ 1° × 1° <sup>d</sup>                                                                                                                                                                      |                                                                                                                                                                                                                                                                                                                                                                                                                                                                                                                                                                                                                                                                                                                                                                                                        | Χ                                                                                                                                                                                                                                                                                                                                                                                                                        |
| WACCM6                                | POP2                                                                                                                                                                                             | MARBL-BEC                                                                                                                                                                                                                                                                                                                                                                                                                                                                                                                                                                                                                                                                                                                                                                                              | • • • •                                                                                                                                                                                                                                                                                                                                                                                                                  |
| $1.25^{\circ} \times 0.9^{\circ}$     | $\sim$ 1° × 1° d                                                                                                                                                                                 |                                                                                                                                                                                                                                                                                                                                                                                                                                                                                                                                                                                                                                                                                                                                                                                                        | $\checkmark$                                                                                                                                                                                                                                                                                                                                                                                                             |
| CAM5                                  | NEMOv3.6                                                                                                                                                                                         | BFMv5.2                                                                                                                                                                                                                                                                                                                                                                                                                                                                                                                                                                                                                                                                                                                                                                                                | • • • •                                                                                                                                                                                                                                                                                                                                                                                                                  |
| $1.25^{\circ} \times 0.9^{\circ}$     | ~1° × 1° °                                                                                                                                                                                       |                                                                                                                                                                                                                                                                                                                                                                                                                                                                                                                                                                                                                                                                                                                                                                                                        | $\checkmark$ , missing for $wo_{60}$                                                                                                                                                                                                                                                                                                                                                                                     |
| ARPEGE-Climat-v6.3                    | NEMOv3.6                                                                                                                                                                                         | PISCESv2-gas                                                                                                                                                                                                                                                                                                                                                                                                                                                                                                                                                                                                                                                                                                                                                                                           | • • • •                                                                                                                                                                                                                                                                                                                                                                                                                  |
| T127 (~150 km)                        | ~1° × 1° °                                                                                                                                                                                       |                                                                                                                                                                                                                                                                                                                                                                                                                                                                                                                                                                                                                                                                                                                                                                                                        | $\checkmark$                                                                                                                                                                                                                                                                                                                                                                                                             |
| IFS 36r4                              | NEMOv3.6                                                                                                                                                                                         | PISCESv2                                                                                                                                                                                                                                                                                                                                                                                                                                                                                                                                                                                                                                                                                                                                                                                               | • •                                                                                                                                                                                                                                                                                                                                                                                                                      |
| T255                                  | ~1° × 1° °                                                                                                                                                                                       |                                                                                                                                                                                                                                                                                                                                                                                                                                                                                                                                                                                                                                                                                                                                                                                                        | Χ                                                                                                                                                                                                                                                                                                                                                                                                                        |
|                                       | UM7.3 Approx. GA1 1.8758° × 1.258°  CanAM5 T63 (~2.8° × 2.8°)  CanAM5 T63 (~2.8° × 2.8°)  CAM6 1.25° × 0.9°  WACCM6 1.25° × 0.9°  CAM5 1.25° × 0.9°  ARPEGE-Climat-v6.3 T127 (~150 km)  IFS 36r4 | UM7.3 Approx. GA1       MOM5 $1.8758^{\circ} \times 1.258^{\circ}$ $\sim 1^{\circ} \times 1^{\circ}$ b         CanAM5       NEMOv3.4.1         T63 ( $\sim 2.8^{\circ} \times 2.8^{\circ}$ ) $\sim 1^{\circ} \times 1^{\circ}$ c         CanAM5       NEMOv3.4.1         T63 ( $\sim 2.8^{\circ} \times 2.8^{\circ}$ ) $\sim 1^{\circ} \times 1^{\circ}$ c         CAM6       POP2 $1.25^{\circ} \times 0.9^{\circ}$ $\sim 1^{\circ} \times 1^{\circ}$ d         WACCM6       POP2 $1.25^{\circ} \times 0.9^{\circ}$ $\sim 1^{\circ} \times 1^{\circ}$ d         CAM5       NEMOv3.6 $1.25^{\circ} \times 0.9^{\circ}$ $\sim 1^{\circ} \times 1^{\circ}$ c         ARPEGE-Climat-v6.3       NEMOv3.6         T127 ( $\sim 150$ km) $\sim 1^{\circ} \times 1^{\circ}$ c         IFS 36r4       NEMOv3.6 | UM7.3 Approx. GA1 MOM5 WOMBAT  1.8758° × 1.258° ~1° × 1° b  CanAM5 NEMOv3.4.1 CMOC  T63 (~2.8° × 2.8°) ~1° × 1° c  CanAM5 NEMOv3.4.1 CanOE  T63 (~2.8° × 2.8°) ~1° × 1° c  CAM6 POP2 MARBL-BEC  1.25° × 0.9° ~1° × 1° d  WACCM6 POP2 MARBL-BEC  1.25° × 0.9° ~1° × 1° d  CAM5 NEMOv3.6 BFMv5.2  1.25° × 0.9° ~1° × 1° c  ARPEGE-Climat-v6.3 NEMOv3.6 PISCESv2-gas  T127 (~150 km) ~1° × 1° c  IFS 36r4 NEMOv3.6 PISCESv2 |

Table 1 (continued)

| Model and reference            | Atmosphere                                     | Ocean                                      | Biogeochemistry                      | Availability                         |
|--------------------------------|------------------------------------------------|--------------------------------------------|--------------------------------------|--------------------------------------|
| GFDL-CM4                       | AM4.0                                          | MOM6 BLINGv2                               |                                      | • •                                  |
| (Held et al., 2019; Dunne      | C96 (~100 km)                                  | $\sim \! 0.25^{\circ} \times 0.25^{\circ}$ |                                      | Χ                                    |
| et al., 2020a)                 |                                                |                                            |                                      |                                      |
| GFDL-ESM4                      | AM4.1 MOM6 COBALTv2                            |                                            | COBALTv2                             | • • • •                              |
| (Dunne et al., 2020b; Stock    | C96 ( $\sim$ 100 km) $\sim$ 0.5° $\times$ 0.5° |                                            | $\checkmark$ , missing for $wo_{60}$ |                                      |
| et al., 2020)                  |                                                |                                            |                                      |                                      |
| IPSL-CM6A-LR                   | LMDZ6A-LR                                      | NEMOv3.6                                   | PISCESv2                             | • • • •                              |
| (Boucher et al., 2020)         | $2.5^{\circ} \times 1.3^{\circ}$               | $\sim$ 1° × 1° °                           | $\sim$ 1° × 1° °                     |                                      |
| MIROC-ES2L                     | CCSR-AGCM                                      | COCO                                       | OECO2                                | • • • •                              |
| (Hajima et al., 2020)          | T42 ( $\sim$ 2.8° $\times$ 2.8°)               | $\sim$ 1° $\times$ 1°                      |                                      | √, missing for MLD                   |
| MPI-ESM1-2-HR                  | ECHAM6.3                                       | MPIOM1.6                                   | HAMOCC6                              | • • • •                              |
| (Müller et al., 2018; Maurit-  | T127 (~100 km)                                 | $0.4^{\circ} \times 0.4^{\circ}$           |                                      | Χ                                    |
| sen et al., 2019)              |                                                |                                            |                                      |                                      |
| MPI-ESM1-2-LR                  | ECHAM6.3                                       | MPIOM1.6                                   | HAMOCC6                              | • • • •                              |
| (Mauritsen et al., 2019)       | T63 (~200 km)                                  | $\sim$ 1.5° $\times$ 1.5°                  |                                      | $\checkmark$                         |
| MRI-ESM2-0                     | MRI-AGCM3.5                                    | MRI-COM4                                   | MRI-COM4                             | •                                    |
| (Yukimoto et al., 2019)        | T159 (~120 km)                                 | ${\sim}1^{\circ} \times 0.5^{\circ}$       |                                      | $\checkmark$ , missing for $wo_{60}$ |
| NorESM2-LM                     | CAM6-Nor                                       | BLOM                                       | iHAMOCC                              | • • • •                              |
| (Tjiputra et al., 2020; Seland | $\sim$ 2 $^{\circ}$ $\times$ 2 $^{\circ}$      | $\sim 1^{\circ} \times 1^{\circ}$          |                                      | $\checkmark$                         |
| et al., 2020)                  |                                                |                                            |                                      |                                      |
| NorESM2-MM                     | CAM6-Nor                                       | BLOM                                       | iHAMOCC                              | • • • •                              |
| (Tjiputra et al., 2020; Seland | $1.25^{\circ} \times 0.9^{\circ}$              | $\sim 1^{\circ} \times 1^{\circ}$          |                                      | Χ                                    |
| et al., 2020)                  |                                                |                                            |                                      |                                      |
| UKESM1-0-LL                    | MetUM-HadGEM3-                                 | NEMOv3.6                                   | MEDUSA-2                             | • • • •                              |
| (Sellar et al., 2019)          | GA7.1                                          | $\sim$ 1° × 1° °                           |                                      | $\checkmark$                         |
|                                | $1.875^{\circ} \times 1.25^{\circ}$            |                                            |                                      |                                      |

 $<sup>^</sup>a$  Resolution is expressed as longitude imes latitude or by the truncation level used for the atmospheric component (symbol  $\sim$  denotes nominal resolution).

<sup>&</sup>lt;sup>b</sup> Higher resolution near the equator (0.338°) and over the Southern Ocean (0.48°).

<sup>&</sup>lt;sup>c</sup> ORCA family tripolar configuration with refinement to 1/3° close to the equator (Madec and Imbard, 1996).

 $<sup>^{\</sup>rm d}\,$  Uniform resolution in the zonal direction (1.125°), varying in the meridional direction.

Table 2. Overview of the variables used.

| Variable name                            | Official name | Unit                               | Description                                                                   |
|------------------------------------------|---------------|------------------------------------|-------------------------------------------------------------------------------|
| Net Primary Production (NPP)             | intpp         | gC m <sup>-2</sup> d <sup>-1</sup> | Vertically integrated total primary (organic carbon) production (includ-      |
|                                          |               |                                    | ing all phytoplankton types)                                                  |
| Equatorward wind-stress $(\tau_v)$       | tauv          | Pa                                 | Downward equatorward wind stress at the surface                               |
| Vertical water velocity (wo)             | WO            | $m s^{-1}$                         | Upward seawater velocity                                                      |
| Nitrate concentration (NO <sub>3</sub> ) | no3           | mol m <sup>-3</sup>                | Mole concentration of NO <sub>3</sub> per unit volume                         |
| Sea Surface Height $(\eta)$              | ZOS           | m                                  | Dynamic sea level above geoid                                                 |
| Mixed Layer Depth (MLD)                  | mlotst        | m                                  | Depth where potential density $\sigma_T$ exceeds the surface value by a fixed |
|                                          |               |                                    | threshold                                                                     |

#### 100 2.2 Model variables

105

110

The impact of climate change on EBUS phytoplankton activity was investigated through changes in NPP, a key indicator of ecosystem functioning and energy flow within marine ecosystems. NPP was vertically integrated over the full water column yet remains representative of upper-ocean layers, as productivity is largely confined to the well-lit shallow layers. In addition to NPP, variations in potential physical and biogeochemical drivers of NPP were also assessed (Table 2). Upwelling-favorable alongshore winds were evaluated using the equatorward and downward component of surface wind stress ( $\tau_v$ ), while upward seawater velocity at 60 m depth ( $wo_{60}$ ) was used as a proxy for upwelling intensity. This particular depth was selected based on studies that have identified it as representative of the typical source layer (Messié et al., 2009; Chavez and Messié, 2009). Although the actual origin of upwelled waters varies in both time and space, backward particle tracking and sensitivity tests on nitrate supply have demonstrated that 60 m provides a reliable estimate.

Nutrient analysis focused primarily on nitrates, as they have been shown to be the dominant limiting nutrient in EBUS (Messié and Chavez, 2015; Bograd et al., 2023; Jacox et al., 2024). NO<sub>3</sub> concentrations were assessed at three depths: the surface, where nutrients are directly consumed by phytoplankton; 60 m, where upwelled waters are supposed to originate from; and 200 m, which characterizes the regional subsurface nutrient reservoir. Variability at this deeper level may strongly influence future NPP trends (Rykaczewski and Dunne, 2010; Jacox et al., 2024).

The sea surface height  $(\eta)$ , used for the geostrophic transport calculation, is the dynamic sea level above geoid, while the ocean mixed layer depth (MLD) is evaluated using the sigma-t  $(\sigma_t)$  criterion, where MLD is defined as the depth at which potential density exceeds the surface value by a model-specific threshold. This metric delineates the base of the well-mixed surface layer and is critical for understanding vertical mixing and nutrient entrainment.

#### 2.3 Data processing

The eastern boundary upwelling systems were delineated using four of the 66 recognized Large Marine Ecosystems (Sherman and Duda, 1999; Sherman, 2015), ocean regions along continental coasts characterized by high primary productivity. In

particular, the definitions of the EBUS spatial domains were associated with their relative oceanographic features, as each system included the associated equatorward surface eastern boundary current. Masks identifying these domains were obtained from the ISIMIP (Inter-Sectoral Impact Model Intercomparison Project 2024), (Inter-Sectoral Impact Model Intercomparison Project (ISIMIP), 2024) repository (Fig. 1).

Climate change impacts on NPP were assessed by computing anomalies, defined for each model as the difference between the projection and the respective values over the baseline period (1985-2014). Ensemble means of both time series and maps of anomalies were calculated, in order to determine dominant patterns across different model outputs. Time series were obtained using spatial averages within each EBUS mask, with weighted means applied to account for differences in grid cell area. Model uncertainty in the projections was evaluated by calculating the standard deviation across the ensemble for each SSP, while scenario uncertainty was assessed using the divergence between SSP trajectories. Spatial anomalies were computed by time averaging over the last 30 years of SSP simulations (2071–2100). Regions of consistent changes across the ensemble were identified, with high model agreement defined as at least 80 % of models aligning on the sign of future changes, as described in Arias et al. (2021).

To investigate the drivers of projected NPP changes, relationships between NPP and its potential drivers were evaluated. Within each EBUS mask, grid cells were classified based on the sign and coherence of anomalies between variable pairs–specifically,  $\tau_v$ –NPP,  $\tau_v$ – $wo_{60}$ ,  $wo_{60}$ –NO<sub>3</sub>, and NO<sub>3</sub>–NPP. For each variable pair, four outcomes exist (both variables increasing, both decreasing, one increasing while the other decreases, and vice versa) with each grid cell categorized accordingly. To ensure that only robust, ensemble-consistent relationships were retained, the classification was applied only if at least 50 % of ensemble members agreed on the relationship. This threshold is lower than that used for standard ensemble anomalies and was preferred over higher thresholds, as sensitivity tests showed that those would have resulted in an excessively large reduction of the agreement area due to the greater number of possible pairwise outcomes. This approach was extended to include three-variable relationships ( $\tau_v$ –NO<sub>3</sub>–NPP), where classifications were only applied if the anomalies of all three variables shared the same sign across the ensemble majority. It should be noted that, while this analysis assesses mechanistically the dynamics of EBUS changes, it does not account for potential non-linearities between variable pairs that may influence the magnitude of changes.

For the analysis of Ekman and geostrophic transports, the methodology outlined in Jacox et al. (2018) has been followed. The zonal Ekman transport, defined as the integrated near-surface transport directed 90° to the right (left) of the surface wind stress in the Northern (Southern) Hemisphere, is computed as:

$$U_{\text{ekm}} = \frac{\tau_v}{\rho_0 f} \tag{1}$$

where  $\rho_0 = 1025 \text{ kg m}^{-3}$  is the seawater density, and f is the Coriolis parameter.

The cross-shore geostrophic flow is calculated as:

$$U_{\text{geo}} = -\frac{g}{f} \frac{\partial \eta}{\partial y} H \tag{2}$$

where g is the gravitational acceleration, and H is the Ekman layer depth, set for simplicity at 30 m, a depth which has been shown to provide a robust approximation (Ding et al., 2021).

**Figure 1.** General ability of the CMIP6 models to simulate observed NPP in the EBUS albeit with certain biases. Time-averaged NPP (1998–2014) from observations (mean between five data products obtained combining ESA OC-CCIv4.1 ocean color observations with five different algorithms) (a), the CMIP6 multi-model mean (b), and the difference between the two (simulated minus observed) (c). The four EBUS are indicated by black borders in each map.

#### 2.4 Model evaluation

Historical observations of global NPP highlight the eastern boundary upwelling systems as highly productive regions (Fig. 1a). NPP estimates for the period 1998–2014 were obtained combining the ocean colour data product from the European Space Agency Ocean Colour Climate Change Initiative project (ESA OC-CCIv4.1) (European Space Agency (ESA), 2024), with five different algorithms: Eppley-VGPM (Eppley, 1971), Behrenfeld-VGPM (Behrenfeld and Falkowski, 1997), Behrenfeld-CbPM (Behrenfeld et al., 2005), Westberry-CbPM (Westberry et al., 2008), and Silsbe-CAFE (Silsbe et al., 2016). Over this historical period, the multi-algorithm mean indicates the highest average NPP in the BenCS, with a mean of about 0.99 gC m<sup>-2</sup>d<sup>-1</sup>, followed by the HumCS with 0.86 gC m<sup>-2</sup>d<sup>-1</sup>, the CanCS with 0.82 gC m<sup>-2</sup>d<sup>-1</sup>, and the CalCS with 0.54 gC m<sup>-2</sup>d<sup>-1</sup>. Productivity patterns show spatial variability, with peak values in localized regions where strong alongshore winds drive deep, nutrient-rich waters to the surface.

Comparison between the observational data and ensemble mean simulations for the same period shows that models capture the large-scale NPP variability and generally identify high productivity regions in EBUS (Fig. 1b), even if with a slight misrepresentation of the precise location of hotspots (especially in CanCS and BenCS). The difference between simulated and observed values (Fig. 1c) reveals some drawbacks, with models underestimating NPP—especially in coastal regions of high productivity. On average, models underestimate NPP by -0.23 gC m<sup>-2</sup>d<sup>-1</sup> in the CalCS, -0.18 gC m<sup>-2</sup>d<sup>-1</sup> in the CanCS, -0.23 gC m<sup>-2</sup>d<sup>-1</sup> in the HumCS, and -0.26 gC m<sup>-2</sup>d<sup>-1</sup> in the BenCS. These discrepancies may be attributed to limitations in model representations of wind dynamics, upwelling process, or biogeochemical parameterizations (e.g. sedimentation).

#### 3 Results

## 140 3.1 NPP projections and associated uncertainties

Projected changes in net primary production under climate change reveal distinct patterns both across and within eastern boundary upwelling systems. An analysis of the multimodel mean time series of NPP, based on a 13 models ensemble (see selection criteria detailed in Materials and Methods), showed that only in the CanCS and HumCS the projected anomalies are non-negligible in magnitude and SSP-consistent in the direction of change (Fig. 2). Comparable results are obtained using the full CMIP6 ensemble (Fig. A1 in the Appendix). By 2100, ensemble mean NPP is anticipated to decline in CanCS (between -7/-10 % under different SSPs), and to increase in HumCS (with larger anomalies for higher emission scenarios, ranging from +2 % under SSP1-2.6 to +7 % under SSP5-8.5). In contrast, ensemble mean anomalies remain close to zero under all SSPs in both CalCS and BenCS. This is due both to the low model agreement on whether NPP will increase or decrease over the entire region, and to the spatial averaging process that masks compensating regional trends within individual models.

We find that model uncertainty across the CMIP6 dominates over scenario uncertainty. While there is often model disagreement on the overall direction of anomalies, for individual members the sign of productivity change remains generally consistent across different SSPs, with larger magnitude of changes for increasing emission pathways (Fig. A2 in Appendix). This suggests that NPP projections are more sensitive to the choice of the model, rather than to the emission scenarios.

Consensus on the sign of anomalies is larger in HumCS and CanCS, with most members projecting respectively NPP increases and decreases, while in the CalCS and BenCS the intermodel spread covers both directions of change. By the end of the 21st century, under SSP5-8.5, intermodel standard deviation is lowest for the HumCS (17 %), followed by the CanCS (26 %), CalCS (28 %), and BenCS (31 %). Across the other scenarios, in CalCS and CanCS intermodel spreads remain comparably large, while in BenCS and HumCS STDs decrease under lower-emissions scenarios.

Although NPP projections at the EBUS scale remain highly uncertain, this is partly due to offsetting positive and negative NPP anomalies within individual EBUS regions, which are often consistently projected across the CMIP6 ensemble. Under SSP5-8.5, ensemble mean NPP is projected to decline in several key regions: in the southern and central coastal area of the CalCS, in the CanCS, and in the equatorward half of BenCS, and at the equatorward limit of the HumCS—where the definition of sub-EBUS is based on the spatial patterns of ensemble-mean NPP anomalies (Fig. 3 for spatial anomalies; Fig. A3 of Appendix for sub-EBUS scale time series anomalies). Strongest NPP declines are consistently projected in the poleward and equatorward regions of CanCS, with relative anomalies reaching -30 % and -20 % respectively. Robust negative changes are also anticipated in equatorward BenCS, going down to -20 %, while in other systems the decline is less pronounced and less robust. In the CalCS, the largest NPP decrease occurs in the historically most productive zone (between 30° N–40° N), peaking at -10 % and being the only high model agreement spot of the whole region. In the Humboldt region, the negative ensemble mean anomalies projected at the equatorward limit are small in magnitude (-2 %) and not robust. Across the rest of the HumCS, positive anomalies are projected (peaking at +16 % around 40° S), with high model agreement between 35°-50° S. Robust positive changes are also projected in the poleward portion of BenCS (up to +10 %), while in the northern and offshore central CalCS the ensemble mean increase (up to +5 %) is not consistent across models.

**Figure 2.** Diverse and uncertain NPP responses across EBUS. Historical (1985-2014, grey) and projected (2015-2100, colored) relative change in NPP for different SSPs using the selected models from CMIP6 ensemble: ensemble means are indicated with solid lines, intermodel standard deviation with shaded regions. Colored vertical bars indicate, for each scenario, the intermodel standard deviation by the end of the century. A 10-year smoothing is applied.

Spatial projections from the full CMIP6 ensemble are comparable to the ones obtained with the 13-model subset (Fig. A4 in the Appendix). Moreover, analysis of NPP anomalies under other SSPs suggests that changes are not scenario-dependent, but rather reflect changes in the underlying processes that drive productivity in these regions (Fig. A5, A6, A7 in Appendix). Spatial patterns of future NPP anomalies show consistent directions of changes across all four SSPs, as the same areas are projected to experience similar productivity trends, with lower emissions pathways showing smaller magnitudes of variation. The level of agreement and the regions of robust changes remain generally the same across scenarios.

## 3.2 Drivers of NPP changes

Projected declines in NPP from the CMIP6 ensemble mean are linked to local weakening of alongshore winds in central CalCS and in the equatorward CanCS and BenCS (Fig. 4). Co-located decreases in  $\tau_v$  and NPP are consistently projected in the central and southern CalCS, in the equatorward Canary and Benguela systems, and in the northern edge of HumCS (28.4 % of total EBUS area), while increases are projected in the poleward sectors of HumCS and BenCS (20.1 %). In the remaining regions the two variables either do not show significant relations (28.4 %), or change in opposite directions, with increasing (decreasing)

**Figure 3.** CMIP6 ensemble mean anomalies of NPP show regionally contrasting trends within individual EBUS, with consistent patterns projected across models. Ensemble mean (13 models) NPP anomalies (2071–2100 relative to 1985–2014) under SSP5-8.5 in the four eastern boundary upwelling systems. Stippling highlights areas of high model agreement. The poleward and equatorward sub-EBUS are indicated in each map by black lines at the latitudes of 30° N (CalCS), 22° N (CanCS), 32° S (HumCS), and 24° S (BenCS).

NPP associated with weaker (stronger)  $\tau_v$  in 9.8 % (13.3 %) of the EBUS area. The influence of seasonality on this finding is limited, as the same analysis obtained for summer data (months of June–August and December–February respectively in the Northern and Southern Hemisphere) results in similar projected relations between NPP and  $\tau_v$  anomalies (Appendix Fig.

190

A8). The main difference is observed in the poleward widening, across all EBUS, of regions of consistently decreasing  $\tau_v$  and NPP, covering 38.7 % of the total EBUS area compared to 28.4 % in the annual case. In the CanCS, summer weakening of  $\tau_v$  extends across the entire system, as the region of increasing  $\tau_v$  shifts further north, outside its boundaries.

Alongshore winds are primary drivers of NPP changes only in the equatorward portions of CanCS and BenCS, and in the central CalCS (Fig. 5). In these areas, models project simultaneous decreases in  $\tau_v$ , surface NO<sub>3</sub>, and NPP, while elsewhere this mechanistic chain connecting coastal wind stress to primary productivity either breaks down or is inconsistent. When the same analysis also includes changes in  $wo_{60}$ , the regions showing consistent declines across all four variables shrink, especially in the CalCS, where they localize around a coastal spot between 30° N–40° N, potentially due to the smaller model availability for the vertical velocity variable (Appendix Fig. A9).

Coincident twenty-first century declines in alongshore winds, seawater vertical velocity, nitrate concentrations, and primary productivity—reflecting the classical upwelling-driven productivity mechanism—are projected in only 25 % of the total EBUS area under SSP5-8.5. This limited overlap highlights the role of additional processes in shaping the NPP response to climate change across these systems.

Upwelling changes are associated with alongshore wind stress only in some sectors of EBUS, as their relationship varies regionally. Upwelling intensity is consistently projected to decrease together with alongshore wind in the equatorward regions of CanCS and BenCS and in the coastal spot of central CalCS, while in the poleward CanCS, both wind strength and upwelling are projected to increase (SSP5-8.5 anomalies of  $\tau_v$  and  $wo_{60}$  are provided respectively in Fig. A10 and A11 of the Appendix). Elsewhere, upwelling intensifies despite a  $\tau_v$  weakening (in the offshore area of CalCS) or no significant relation is projected between the two variables (in the poleward areas of BenCS and HumCS).

Moreover, while the upwelling weakening is associated with future declines in surface nitrates, regions of future  $wo_{60}$  intensification are not showing corresponding NO<sub>3</sub> changes. In the equatorward areas of CanCS, HumCS and BenCS and in the central CalCS negative  $wo_{60}$  anomalies are consistently associated with reduced surface nitrates (36 % of total EBUS area). Elsewhere, despite the intensified upwelling, the CMIP6 ensemble anticipates a general surface NO<sub>3</sub> decline (31 % of total EBUS area).

Finally, while NPP and surface NO<sub>3</sub> are consistently projected to decrease together in some EBUS areas, positive NPP anomalies are not linked to NO<sub>3</sub> changes. The nitrate decrease is consistently associated with reduced NPP everywhere except in the offshore and northern California system, where model agreement is low (in 46 % of total CalCS area), and in the poleward sector of BenCS and across most of HumCS, where NPP increases (where this relation is consistent in 16 % of the BenCS and in 63 % of HumCS areas, respectively).

#### 4 Discussion

## 4.1 NPP changes and the role of upwelling-favorable winds

Projected directions and magnitudes of NPP changes in the EBUS are broadly in line with earlier findings from CMIP5 (Cabré et al., 2015) and CMIP6 ensembles analysis (Bograd et al., 2023; Kwiatkowski et al., 2020). Signs of changes are

Figure 4. Variable relationships between projected anomalies in  $\tau_v$  and NPP within and across EBUS. Signs of twenty-first century colocalized anomalies in  $\tau_v$  and NPP under SSP5-8.5 (2071-2100). Grid cells are colored if there is  $\geq 50$  % model agreement.

mostly coherent for single models across SSPs, indicative of low scenario sensitivity, but are not always consistent across the ensemble. For the BenCS and CalCS, models do not agree on the sign of change by the end of the twenty-first century, while in the CanCS and HumCS, projected trends are respectively towards lower and higher productivity, results generally aligning with past studies (Bograd et al., 2023). Uncertainties of NPP projections in the EBUS are large, since the usual challenges in

Figure 5. Consistent decreases of  $\tau_v$ , surface NO<sub>3</sub>, and NPP in the equatorward portions of CanCS and BenCS and in central CalCS, variable relationships elsewhere. Signs of twenty-first century co-localized anomalies among key variables under SSP5-8.5 (2071–2100). Grid cells are colored if there is  $\geq 50$  % model agreement on the relationship between anomalies of:  $\tau_v$  vs.  $wo_{60}$  (first column),  $wo_{60}$  vs. surface NO<sub>3</sub> (second), surface NO<sub>3</sub> vs. NPP (third), and  $\tau_v$ , surface NO<sub>3</sub>, and NPP combined (fourth).

capturing the full range of influencing processes, both physical and biogeochemical (Frölicher et al., 2016; Tagliabue et al., 2021), are exacerbated by spatially varying, and potentially offsetting, directions of anomalies across the systems.

The projected NPP declines in the equatorward regions of CanCS and BenCS and in the coastal area of central CalCS are consistent with a weakening of upwelling-favorable winds, while, outside of these regions, contributions from other factors dominate. While early theories had anticipated a uniform intensification of upwelling due to global warming (Bakun, 1990), the CMIP6 ensemble analysis projects an increase (decrease) of  $\tau_v$  in the poleward (equatorward) areas of the EBUS (Rykaczewski et al., 2015). The future wind stress weakening in the equatorward sectors of CanCS and BenCS and in central CalCS induces a corresponding reduction of upwelling intensity, thereby leading to smaller nitrates supply and primary productivity (Fig. 5).

## 4.2 Geostrophic transport and wind-stress curl may affect future upwelling

While CMIP6 models consistently indicate that weaker alongshore winds result in reduced productivity, their strengthening does not directly induce higher NPP (Fig. 4, 5). Upwelling is primarily driven by alongshore wind stress in central CalCS and in the equatorward areas of CanCS and BenCS (decreasing  $\tau_v$  and  $wo_{60}$ ), and in the poleward area of the CanCS (increasing  $\tau_v$  and  $wo_{60}$ ). Outside these regions,  $\tau_v$  and  $wo_{60}$  show either inconsistent relations (in the poleward BenCS and HumCS) or opposite directions of changes ( $wo_{60}$  intensifies despite  $\tau_v$  weakening around the coastal CalCS area between 30° N–40° N). This indicates that vertical water velocity is influenced by additional physical mechanisms beyond the alongshore wind forcing. Geostrophic transport changes can modulate upwelling by counteracting wind-driven changes in certain EBUS areas (Appendix Fig. A12). Consistent with previous studies (Jacox et al., 2018), we find that historical ensemble mean Ekman and geostrophic transport generally oppose each other (Appendix Fig. A12a), while projected ensemble mean anomalies of the two reinforce each other across most sub-EBUS domains (Appendix Fig. A12b). However, in some regions—such as the poleward BenCS—the projected ensemble mean changes diverge, offsetting each other, as shown also by Jing et al. (2023) and Du et al. (2024). Moreover, the mismatch between projections of decreasing  $\tau_v$  and increasing  $wo_{60}$  around the localized spot between 30° N–40° N of the CalCS aligns with the potential role of wind-stress curl in determining offshore upwelling (>50 km from the coast) (Bograd et al., 2023; Jacox et al., 2014).

## 4.3 Stratification and subsurface nutrients modulate nutrient supply

Projected twenty-first century changes in upwelling intensity are insufficient to explain surface ocean NO<sub>3</sub> anomalies in the EBUS. Regions of enhanced upwelling intensity do not consistently show corresponding surface nitrate increases, as upper ocean nutrient supply is also impacted by perturbations to water column stratification and subsurface nutrient reservoirs.

In the poleward portion of the CanCS, increased stratification and a shallower mixed layer depth likely drive NPP decline despite intensified upwelling (Fig. 5). Although upwelling of cold, deep waters typically offsets warming effects in EBUS (García-Reyes et al., 2015), a consistent MLD shoaling is projected everywhere with the exception of local HumCS regions. MLD shoaling is particularly pronounced in the poleward CanCS (Fig. 6). The MLD in this region is historically deeper ( $\sim$ 60 m) than in other EBUS and is expected to shoal by substantially more ( $\sim$ 20 m). Under such conditions, enhanced stratification may overcompensate upwelling intensification, reducing vertical mixing and surface nutrient supply, and lowering productivity. Similar trends have been projected off the nearby Iberian Peninsula (Sousa et al., 2020), reinforcing this finding. Shoaling of upwelling source depth has also been linked with increased stratification, potentially resulting in waters with lower nutrient concentrations reaching the surface (Jacox and Edwards, 2011), however water parcel trajectory analysis was beyond the scope of this study.

In addition to enhanced stratification, simulated changes in subsurface nitrate reservoirs may be as important as changes in upwelling intensity in determining EBUS NPP projections. Euphotic zone nutrient supply is a consequence of both water mass transport and the concentration of nutrients associated with those waters. The impact of subsurface nitrate anomalies on NPP is limited because perturbations do not consistently propagate to shallower waters, and because, where they do, NO<sub>3</sub> availability

**Figure 6.** Strong mixed layer depth shoaling in the poleward CanCS. Ensemble mean (12 models) MLD anomalies (2071–2100 relative to 1985–2014) under SSP5-8.5 in the Canary EBUS. Stippling highlights areas of high model agreement.

may not be the primary driver of NPP variability (Echevin et al., 2008). Consistent declines in 200 m NO<sub>3</sub> and NPP are projected in central coastal California and in equatorward CanCS and BenCS (Appendix Fig. A13). Reduced nitrate throughout the water column contributes to the NPP decrease, reinforcing the effect of weker wind stress and upwelling intensity. Surface nitrate reductions may also be associated with the impoverishment of upwelled waters in the offshore California system (Appendix Fig. A14), however impacts on NPP are uncertain. While nitrates are projected to increase around 200 m in this region (Appendix Fig. A15), these waters do not reach the typical source depth (60 m), where anomalies are negative (Appendix Fig. A16), resulting in reduced nutrient supply despite enhanced upwelling. Contrary to earlier findings indicating that source water nutrient content is the dominant driver of productivity in the CalCS (Rykaczewski and Dunne, 2010; Jacox et al., 2024), the CMIP6 ensemble shows inconsistent links between source water NO<sub>3</sub> concentrations and NPP, potentially due to the greater model ensemble size. In poleward HumCS and BenCS, lower nitrate supply is primarily driven by nutrient-poor upwelling, but the effect on NPP appears limited (Appendix Fig. A13). This could be due to historically low nitrate limitation in these areas, with future productivity more influenced by the relaxation of light limitation associated with the shoaling of the mixed layer,

**Figure 7.** Evidence that subsurface nitrate increases enhance projected NPP in the CanCS. IPSL-CM6A-LR projected anomalies (2071–2100 relative to 1985–2014) of NPP (a) and 200 m NO<sub>3</sub> concentrations (b) under SSP5-8.5 in the Canary EBUS.

an effect highlighted in studies on seasonal NPP variability (Echevin et al., 2008; Vergara et al., 2017). It should be noted that while this study focuses on bottom-up control associated with nitrate supply to the euphotic zone, additional factors may shape future EBUS productivity. Specifically, co-limitation by nutrients such as iron (Browning and Moore, 2023; Echevin et al., 2008; Chavez and Messié, 2009), or top-down control by zooplankton grazing (Calbet and Landry, 2004; Slaughter et al., 2006) could be relevant drivers of NPP.

We note that subsurface nitrate anomalies remain a key factor influencing NPP projections for individual models. For example, IPSL-CM6A-LR projects a relative NPP increase of more than 50 % by the end of the century over the Canary EBUS region north of 20° N (Fig. 7a), diverging from the ensemble mean, which projects a NPP decrease of about 7 % in this region. This divergence can be attributed to an increase in subsurface NO<sub>3</sub> concentrations in the IPSL model (mean anomaly of about 6.2 mmol m<sup>-3</sup>, Fig. 7b), which is more than double the ensemble mean (approximately 2.4 mmol m<sup>-3</sup>). This relatively high nitrate increase could be linked to nitrogen fixation, which has been shown to substantially increase in this model (Bopp et al., 2021). Enhanced diazotrophy in oligotrophic regions outside of the Canary EBUS, could lead to higher subsurface nitrate concentrations inside the system due to nutrient-rich water transport at depth.

## 5 Conclusions and Perspectives

The uncertainties associated with CMIP6 NPP projections in the four major EBUS stem from the complex interplay of multiple factors. This study highlights that the traditional paradigm of wind-driven upwelling perturbations determining future NPP changes only holds in 25 % of EBUS areal extent (in central CalCS and in equatorward portions of the CanCS and BenCS). Elsewhere, changes in additional mechanisms, both physical—geostrophic transport, wind-stress curl, stratification—and biogeochemical—subsurface nutrient reservoirs—are required to explain NPP responses in these ecologically and economically highly productive systems.

Assessing future NPP is complicated by the diverse spatial and temporal scales characterizing the underlying processes, which may exhibit non-monotonic behaviors or delayed responses (Jacox et al., 2016). While the atmosphere rapidly reacts to climate forcing, ocean inertia causes delayed changes in seawater properties. Natural variability, especially low-frequency modes such as El Niño-Southern Oscillation or the Pacific Decadal Oscillation, may influence EBUS NPP, hindering the ability to distinguish anthropogenic impacts (Sydeman et al., 2014; Jacox et al., 2015). Moreover, under projections beyond the typical end-of-century horizon, basin-scale feedbacks may alter properties of source waters feeding upwelling systems (e.g. through nutrient trapping (Moore et al., 2018)). In the EBUS, the effects of such changes are potentially further delayed, as the mean age of source waters can be decades to centuries (Resplandy et al., 2013; Bakun et al., 2015).

Although the CMIP6 Earth System Models used in this study offer a large-scale framework for assessing changes in EBUS NPP, their relatively low spatial resolution introduces certain limitations. ESMs may poorly represent coastal upwelling and fine-scale features such as mesoscale eddies or coastal trapped waves (Sylla et al., 2022; Small et al., 2015; Chang et al., 2023; Jing et al., 2023). Nevertheless, these global models permit the simulation of large-scale ocean and atmosphere dynamics at feasible computational cost. Given that the CMIP6 models indicate that large-scale processes extending beyond EBUS regions can influence local biogeochemical conditions, it is important that EBUS-specific regional ocean models are forced with boundary conditions derived from such global models to ensure consistency.

*Data availability.* The Earth System Model output used in this study is available via the Earth System Grid Federation. https://esgf-node.ipsl.upmc.fr/projects/esgf-ipsl/.

Author contributions. Conceptualization: EC, LK, LB; Methodology: EC, LK, LB; Investigation: EC, LK, LB; Visualization: EC; Writing—original draft: EC; Writing—review & editing: EC, LK, LB.

Competing interests. The authors declare no conflict of interests.

## Appendix A

**Figure A1.** NPP projections using the full CMIP6 ensemble. Historical (1985-2014, grey) and projected (2015-2100, colored) relative change in NPP for different SSPs using the full CMIP6 ensemble (18 models for SSP5-8.5, 17 for SSP2-4.5, 15 for SSP3-7.0 and SSP1-2.6): ensemble means are indicated with solid lines, intermodel standard deviation with shaded regions. Colored vertical bars indicate, for each scenario, the intermodel standard deviation by the end of the century. A 10-years smoothing is applied.

**Figure A2.** Low scenario sensitivity of NPP changes. For each CMIP6 model, each point represents the NPP anomaly, averaged over the final 30 years of simulations (2071-2100) and spatially averaged across the entire system domains, under the four SSPs.

**Figure A3.** NPP projections in the sub-EBUS. Historical (1985-2014, grey) and projected (2015-2100, colored) relative change in NPP for different SSPs using the selected models from CMIP6 ensemble: ensemble means are indicated with solid lines, intermodel standard deviation with shaded regions. Colored vertical bars indicate, for each scenario, the intermodel standard deviation by the end of the century. A 10-year smoothing is applied.

**Figure A4.** NPP anomalies in the full CMIP6 ensemble are comparable to projections from the selected ensemble. Ensemble mean (18 models) NPP anomalies (2071–2100 relative to 1985–2014) under SSP5-8.5 in the four eastern boundary upwelling systems. Stippling highlights areas of high model agreement.

**Figure A5.** Consistent directions of NPP changes across SSPs. Ensemble mean NPP anomalies (2071–2100 relative to 1985–2014) under SSP1-2.6 (15 models) in the four eastern boundary upwelling systems. Stippling highlights areas of high model agreement.

**Figure A6.** Consistent directions of NPP changes across SSPs. Ensemble mean NPP anomalies (2071–2100 relative to 1985–2014) under SSP2-4.5 (17 models) in the four eastern boundary upwelling systems. Stippling highlights areas of high model agreement.

**Figure A7.** Consistent directions of NPP changes across SSPs. Ensemble mean NPP anomalies (2071–2100 relative to 1985–2014) under SSP3-7.0 (15 models) in the four eastern boundary upwelling systems. Stippling highlights areas of high model agreement.

Figure A8. Summertime relationships between projected anomalies in  $\tau_v$  and NPP are similar to yearly relationships. Signs of twenty-first century co-localized anomalies in summer  $\tau_v$  and summer NPP under SSP5-8.5 (2071-2100). Grid cells are colored if there is  $\geq 50$  % model agreement.

Figure A9. Consistent decreases of  $\tau_v$ ,  $wo_{60}$ , surface NO<sub>3</sub>, and NPP only in the equatorward portions of CanCS and BenCS and in a localized spot of CalCS. Signs of twenty-first century co-localized anomalies among key variables under SSP5-8.5 (2071–2100). Grid cells are colored if there is  $\geq 50$  % model agreement on the relationship between anomalies of  $\tau_v$ ,  $wo_{60}$ , surface NO<sub>3</sub>, and NPP combined.

Figure A10. Meridional gradient of  $\tau_v$  anomalies, strengthening poleward and weakening equatorward. Ensemble mean (13 models)  $\tau_v$  anomalies (2071–2100 relative to 1985–2014) under SSP5-8.5 in the four eastern boundary upwelling systems. Stippling highlights areas of high model agreement.

**Figure A11.** Projected changes in upward sea water velocity at 60 m. Ensemble mean (10 models)  $wo_{60}$  anomalies (2071–2100 relative to 1985–2014) under SSP5-8.5 in the four eastern boundary upwelling systems. Stippling highlights areas of high model agreement.

**Figure A12.** Relative contributions of Ekman and geostrophic transports in EBUS. Ensemble mean (13 models) Ekman and geostrophic transports in the sub-EBUS computed according to equations (1) and (2) respectively, for historical (1985-2014) (a), and anomalies (2071–2100 relative to 1985–2014) (b).

Figure A13. NPP and 200 m  $NO_3$  change accordingly only in the equatorward areas of CanCS and BenCS and in the coastal spot between 30° N–40° N of CalCS. Signs of twenty-first century co-localized anomalies in  $NO_3$  and NPP under SSP5-8.5 (2071-2100). Grid cells are colored if there is  $\geq 50$  % model agreement.

**Figure A14.** Ensemble mean (13 models) anomalies (2071–2100 relative to 1985–2014) of NO<sub>3</sub> concentrations at surface under SSP5-8.5 in the four eastern boundary upwelling systems. Stippling highlights areas of high model agreement.

**Figure A15.** Ensemble mean (13 models) anomalies (2071–2100 relative to 1985–2014) of NO<sub>3</sub> concentrations at 200 m depth under SSP5-8.5 in the four eastern boundary upwelling systems. Stippling highlights areas of high model agreement.

**Figure A16.** Ensemble mean (13 models) anomalies (2071–2100 relative to 1985–2014) of NO<sub>3</sub> concentrations at 60 m depth under SSP5-8.5 in the four eastern boundary upwelling systems. Stippling highlights areas of high model agreement.

Acknowledgements. We acknowledge the World Climate Research Programme's Working Group on Coupled Modelling, which is responsi320 ble for CMIP. For CMIP, the US Department of Energy's Program for Climate Model Diagnosis and Intercomparison provided coordinating
support and led the development of software infrastructure in partnership with the Global Organisation for Earth System Science Portals. The
authors also thank the IPSL modelling group for the software infrastructure, which facilitated CMIP analysis. This study benefited from the
ESPRI (Ensemble de Services Pour la Recherche l'IPSL) computing and data center (https://mesocentre.ipsl.fr) which is supported by CNRS,
Sorbonne Université, Ecole Polytechnique, and CNES and through national and international grants. We are very grateful to Olivier Torres

for his assistance in data management. Researchers received funding from the CArbon Losses in Plants, Soils and Ocean (CALIPSO) project
funded through the generosity of Eric and Wendy Schmidt by recommendation of the Schmidt Sciences programme (L.B.) and the project
TipESM "Exploring Tipping Points and Their Impacts Using Earth System Models" (funded by the European Union. Grant Agreement No:
101137673) (L.K.).

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
