# Peer review of "Beyond wind-induced upwelling: diverse drivers of future productivity in eastern boundary upwelling systems"

_EGUsphere, 2025_

## Referee Comment (RC1)

This is an interesting study in which the authors use a suite of ESMs to evaluate future changes in NPP and their associated drivers in the EBUSs. The study highlights that the traditionally assumed key mechanism, upwelling-favorable winds, can only explain about 25% of NPP changes in the EBUS, while the remaining three-quarters are influenced by diverse drivers.

However, I do not recommend this manuscript for publication at this stage, as major revision is required. In my view, the current analysis of the relationship between upwelling-favorable winds and future NPP changes is somewhat simplistic and would benefit from a deeper investigation of model biases, inter-model differences, and regional variability. In addition, the discussion of the "diverse drivers" is uneven: analyses of physical processes such as geostrophic transport and wind-stress curl are extremely limited, whereas stratification and biogeochemical control of subsurface nutrient reservoirs are discussed in much greater detail. I therefore suggest that the authors thoroughly reconsider and improve the structure and balance of the manuscript.

Major Comments

1. Rearrangement of the Introduction section. In L53, the authors state "Alongside projected changes in vertical velocity," which may not be fully appropriate, as both changes in vertical velocity and in nutrient supply to the surface are discussed previously. Moreover, the subsequent paragraph focuses on the timing of the upwelling season, which could also be classified as changes in vertical velocity. I therefore suggest reorganizing the Introduction between L32-58. Specifically, the authors could first introduce the projected changes in different mechanisms that modulate the upwelling intensity and duration, and then present additional sources influencing nutrient supply to the surface beyond the physical drivers. L59 will follow just after this, summarizing how variations in upwelling itself and changes in biogeochemical properties of source waters together modulate nutrient supply in EBUSs. such a reorganization could improve the logical flow of the intro.

2. As the manuscript highlights regional differences in Fig. 3, additional evaluation would be valuable. For example, as shown in Fig. A2, most models project a weak positive change

in the CalCS, while fewer models project a strong negative change, particularly CanESM5. It would be useful to clarify whether CanESM5 exhibits mean NPP anomalies comparable to those of other models, or whether there are substantial spatial differences. Moreover, could these differences be related to model biases? For instance, does CanESM5 already exhibit biases in simulating NPP in the CalCS during the historical period?

In addition, the model agreement showed in Fig. 3 raises further questions. In the CalCS, model agreement is weak and the projections remain highly variable, which is acceptable; and in the CanCS, model agreement is also modest, yet the projections show relatively high consensus. The high uncertainty in the BCS can be explained by Fig. A3, where the poleward and equatorward regions show relatively consistent changes but in opposite directions. I believe adding this information to this section would be better.

Therefore, the diversity of NPP projections warrants further evaluation to identify and discuss the potential sources of uncertainty that lead to this strong sensitivity to model choice, such as model biases, inter-model differences, or spatial heterogeneity.

3. Some rearrangement of the Discussion section is recommended. I found that Section 4.1 is somewhat repetitive, covering material similar to the Introduction. Some of the content could be interspersed into the preceding text. This section also does not present any additional analysis. I therefore suggest deleting Section 4.1 and incorporating any necessary statements into Sections 3 and 5.

Section 4.2 is somewhat unclear and abrupt. It appears to discuss the mismatch between wo60 and tau, possibly attributing it to contributions from wind stress curl, but geostrophic transport is also mentioned without clear connection. I suggest integrating the discussion of wind stress curl, together with Figure A10, into Section 3.2 for a clearer and more coherent presentation. Additionally, the role of geostrophic transport requires further explanation.

4. The manuscript includes many different regions and numerous subplots that are frequently cross-referenced in the text. However, in several places the relevant figures or panels are not explicitly specified, making it difficult for readers to follow. I recommend labeling multi-panel figures with clear identifiers (such as Fig. 5; e.g., a, b, c, d) and explicitly referring to these labels in the text to improve clarity and readability.

Detail Comments:

1. There are some errors in the line numbers on page 6 and 7. A large section of text between lines 115 and 120 is missing line numbers.

2. L25-29 should include appropriate references not only for the "*other physical processes*", but also for the broader statements made in this part of the text.

3. In L67-74, the influence of decadal and multidecadal climate variability on NPP should be introduced in greater depth to reinforce the importance of the limited duration of the observational records.

4. L105: The authors use the variables "*tauu/tauv*" to represent the surface wind stress. These variables originate from the atmospheric model on its native grid, which differs from the ocean model grid. Could the authors please clarify whether these fields were first interpolated onto the ocean grid and then regridded to the common 1-degree resolution, or whether they were directly regridded from the atmospheric grid to the 1-degree grid? This distinction is important because wind stress is interpolated to the ocean grid to drive coastal upwelling, and CMIP also provides the "*tauuo/tauvo*" variables on the ocean grid.

5. What is the relative width (200 km?) of the LME mask?

6. L157 Stating the exact number of models projecting positive versus negative changes would provide useful quantitative information.

7. L207-208 Please specify which figure is being referred to, the second left column in Fig 5?

8. L272 "While nitrates are projected to increase around 200 m in this region (Appendix Fig. A15), these waters do not reach the typical source depth (60 m)" can this be attributed to enhanced stratification?